# Phenology and Potential Fecundity of *Neoleucopis kartliana* in Greece

**DOI:** 10.3390/insects13020143

**Published:** 2022-01-28

**Authors:** Nikoleta Eleftheriadou, Umar Lubanga, Greg Lefoe, M. Lukas Seehausen, Marc Kenis, Nickolas G. Kavallieratos, Dimitrios N. Avtzis

**Affiliations:** 1Forest Research Institute—Hellenic Agricultural Organization Demeter, 57006 Thessaloniki, Greece; dimitrios.avtzis@fri.gr; 2Department of Jobs, Precincts and Regions, Invertebrate & Weed Sciences, Agriculture Victoria Research Division, AgriBio Centre, Bundoora, VIC 3083, Australia; umar.lubanga@agriculture.vic.gov.au (U.L.); Greg.Lefoe@agriculture.vic.gov.au (G.L.); 3Centre for Agriculture and Bioscience International, Rue des Grillons 1, 2800 Delémont, Switzerland; l.seehausen@cabi.org (M.L.S.); m.kenis@cabi.org (M.K.); 4Laboratory of Agricultural Zoology and Entomology, Faculty of Crop Science, Agricultural University of Athens, 75 Iera Odos Str., 11855 Athens, Greece; nick_kaval@aua.gr

**Keywords:** Chamaemyiidae, Margarodidae, voltinism, egg development

## Abstract

**Simple Summary:**

The silver fly *Neoleucopis kartliana* Tanasijtshuk (Diptera, Chamaemyiidae) is the most abundant predator of the giant pine scale (GPS), *Marchalina hellenica* (Hemiptera, Margarodidae), and is considered a major factor in controlling GPS populations in Greece and Turkey. GPS has recently been detected in Australia. While generally not harmful to trees in its area of origin, GPS has a detrimental impact on pine trees in Australia and, therefore, needs to be controlled. As part of an evaluation of the silver fly for importation to Australia where it may be used as a biological control agent against GPS, we studied several aspects of the fly’s life history, namely its seasonal occurrence and number of generations per year (phenology), its acceptance of artificial food sources as adult flies, and the number of eggs females produce over their lifetime. We found that the fly has three generations per year and feeds on all life stages of GPS (eggs, nymphs, and adults). Adults readily feed on a mixture of sugar and dry yeast, and females emerge with no or few eggs and develop more as they age.

**Abstract:**

*Neoleucopis kartliana* Tanasijtshuk (Diptera, Chamaemyiidae) is the most abundant predator of the giant pine scale (GPS), *Marchalina hellenica* (Hemiptera, Margarodidae) in Greece. GPS is native to Greece and Turkey, where it is not considered a pest of *Pinus* spp., but a valuable resource for pine honey production. However, its introduction to new areas leads to high population densities of the scale, linked to declines in tree health and insect biodiversity. To assess the potential use of *N. kartliana* for a classical biological control program in Australia, we studied selected life-history traits of the silver fly, namely its phenology in northern Greece, feeding preferences of adult flies on artificial food sources, and potential fecundity of female flies. The silver fly was present in every site in northern Greece studied and was found to have at least three generations per year in this area. The fly’s overall sex ratio was 1:1, and adult females emerged with no or few mature eggs in their ovaries, but egg production was exponential until at least the eighth day after emergence. These findings increase our knowledge about the biology of *N. kartliana* and aided in the evaluation of the silver fly as a classical biological control agent against invasive GPS in Australia.

## 1. Introduction

The family Chamaemyiidae (Diptera) is a group of small flies, commonly known as silver flies, whose larvae prey on sternorrhynchous Hemiptera, particularly adelgids, aphids, mealybugs, scales [1,2], and psyllids [3]. The majority of chamaemyiids have one to three generations [4] and inhabit grassland, lowland, and montane habitats [5]. Larvae feed on adult soft-bodied hemipteran species, as well as on their nymphs and eggs [6], and they either pupariate on twigs and branches where their prey is found [7] or drop from the tree and pupariate in the soil [8]. Regarding the oviposition in Diptera, factors such as environmental temperature [9], quality and quantity of larval [10] and adult diet [11,12,13], mating [14], adult population density [9,12], age [13,15], photoperiod [16], and relative humidity [17] affect female egg production. Several Chamaemyiidae species have been successfully utilized in classical biological control programs throughout the world, e.g., Hawaii [18], New Zealand [19], and Chile [20,21]. Despite the potential of Chamaemyiidae as biological control agents against soft-bodied hemipteran species, the family has been understudied, and the biology and morphology of many species are not adequately described [1].

*Neoleucopis kartliana* Tanasijtshuk (Diptera, Chamaemyiidae) has been successfully used as a biological control agent against giant pine scale (GPS), *Marchalina hellenica* (Gennadius) (Hemiptera, Margarodidae) on the island of Ischia (Italy) [22]. GPS is a univoltine sap-sucking insect native to the eastern Mediterranean region, particularly Greece and Turkey. The scale feeds on *Pinus* spp., especially *P. brutia* and *P. halepensis*, but it can also infest *Abies cephalonica* Loudon (Pinales: Pinaceae) [23]. In its native range, it is considered an economically important insect for the apiculture industry rather than a major pest of *Pinus* spp., since it rarely causes tree mortality [24,25]. GPS excretes a sweet, glutinous substance called honeydew, which is collected and converted by bees into pine honey and represents 60–65% of the annual honey production in Greece [23,26]. Due to its importance to apiculture, GPS has been deliberately introduced to new areas of Greece and to the Italian island of Ischia [27], where, on several occasions, it became a pest, reaching high population densities associated with the decline in tree health and reduction in insect biodiversity on pines [25]. In late 2014, GPS was detected in Australia (Melbourne and Adelaide) on a novel host: the North American species *Pinus radiata* D. Don (Pinaceae), which represents 74.5% of the nation’s softwood plantation estate [28]. Since its discovery, GPS population densities have increased dramatically, causing significant damage to untreated *P. radiata* in urban and peri-urban settings and threatening the pine forest industry of Australia [29]. The combination of GPS invading a novel environment without its natural enemies and the availability of suitable host trees increases the likelihood of GPS damaging susceptible trees and plantations if not controlled.

Recent research on the scale’s natural enemy complex has shown that the silver fly *N. kartliana* is the most abundant predator among the natural enemies of GPS in its native range [29,30], suggesting the potential of *N. kartliana* as a classical biological control agent in Australia [29,31]. The species was previously studied by Gaimari et al. [32], who presented an extensive description of the morphology and biology of the silver fly in southern Greece. Here, we add to the knowledge about the species by (1) investigating the phenology of *N. kartliana* in northern Greece and (2) presenting novel data on the egg development in female flies (egg load or potential fecundity).

## 2. Materials and Methods

To study the phenology and occurrence of *N. kartliana* in northern Greece, we collected GPS-infested pine tree twigs and branches every 7–10 days between 6 November 2019 and 21 October 2021 from Kedrinos Lofos in Thessaloniki (57 sampling repetitions). No sampling took place between 4 March and 20 May 2020 because of the closure of the laboratory due to the COVID-19 pandemic. Additionally, to investigate the presence of the fly in different regions, we collected infested twigs and branches from eight sites in northern Greece: Stratoni-Stratoniki, Parthenonas, Katerini-Makriyalos, Pyrghetos-Tempi, Edessa (2–3 sampling repetitions each), Arnea, Alexandroupoli, and Thassos (1 sampling repetition each) (Figure 1). The samples were then transferred to the Laboratory of Forest Entomology (Forest Research Institute, Hellenic Agricultural Organization Demeter) at Vassilika (Thessaloniki, Greece), where random samples of GPS-infested twigs were examined under a stereoscope Zeiss Stemi 508 (Zeiss, Oberkochen, Germany, magnification range 6.3–50×) to determine and count all silver fly and GPS stages present on the twig. The silver fly’s developmental stages were evaluated according to the descriptions of Gaimari et al. [32] and the life stage of GPS according to those of Hodgson and Gounari [33]. Additionally, any species found on the infested branches were collected and kept in ethanol for future identification to further contribute to the description of the natural enemy complex of GPS. For a graphical analysis of GPS and *N. kartliana* phenology, the number of individuals per developmental stage and species was calculated as a percentage relative to other stages (*n* = minimum 100 for GPS).

For the investigation of selected life-history traits of adult *N. kartliana*, all remaining GPS-infested branches potentially containing *N. kartliana* were transferred to well-ventilated cages (60 × 60 × 60 cm) that were placed inside a climate chamber Termaks KB8400F (Termaks, Bergen, Norway) set to 23 °C and 60% relative humidity. In order to resemble the conditions from dusk to dawn, the climate chamber had a 16:8 h light:dark photoperiod with a gradual transition (lasting one hour) from 0% light to 100% light, and vice versa. The cages were inspected every 1–2 days in search of any *N. kartliana* adults.

To determine the overall sex ratio, emerging *N. kartliana* adults were individually collected in small falcon tubes (5.5 cm length and 1.5 cm diameter), and their sex was identified by visual inspection of their genitalia according to the descriptions provided by Gaimari et. al. [32] using a stereoscope (Zeiss Stemi 508, magnification range 6.3–50×).

To study the acceptance of artificial food sources as substitutes for GPS honeydew, which was presumed to be their natural food source [30] in the manner of other Chamaemyiid species [34], adults (*n* = 270) were gradually transferred to smaller cages (30 × 30 × 30 cm, mean number of adults per cage 15 ± 5) between 12 August and 5 October 2020. These individuals were provided with water (through a constantly soaked cloth strip laid loosely on a vial) and five different media simultaneously: (1) pine honey; (2) pine honey mixed with dry yeast diluted in water (2 mL:1 gr:100 mL); (3) water-diluted pasteurized milk (50:50) provided through soaked cotton on a petri dish (8.5 cm diameter); (4) dry yeast diluted with sugar; and (5) raw, moist yeast mixed with sugar. Artificial food sources (4) and (5) were both provided in different rates (5–50%) and different liquidity states on cotton laid over petri dishes. All food sources were renewed every 2–3 days, and the behavior of the flies was observed twice per day (morning and noon) every 1–2 days. Cotton was used as a substrate for all artificial food sources to resemble the cotton-like wax excreted by GPS under which the honeydew is naturally produced.

To investigate the development of eggs in the ovaries of adult *N. kartliana* females over time (often called egg load or potential fecundity), branch samples from Kedrinos Lofos (Thessaloniki) were placed in cages and positioned near a natural light source for at least two hours. Thus, emerging flies were attracted to the light source, promptly collected in small falcon tubes (5.5 cm length and 1.5 cm diameter), and isolated in small containers (7 cm height and 5 cm diameter) in which they were provided with water and artificial food source (4) (see paragraph above for more details on the artificial food sources). The containers were placed in a climate chamber with the conditions as described above for infested branches. After 3, 6, or 8 days of rearing, females were killed by placing them into 99% ethanol for several minutes. Flies were then dissected under a microscope, and the eggs were counted either immediately after emergence (*n* = 25), or 3 (*n* = 35), 6 (*n* = 34), and 8 (*n* = 38) days after emergence. To be considered mature, eggs had to carry the stripe pattern typically visible on oviposited eggs [32], which was visible at 40× magnification, confirming that the eggshell was fully developed.

The influence of female age on the number of mature eggs in the ovary was analyzed using a negative binomial generalized linear model fitted with the *nb.glm* function of the MASS package [35] in R [36]. Female age was taken as a continuous independent variable and the number of eggs as the dependent variable. The Poisson distribution was not used because the residuals were overdispersed, as indicated by Pearson’s chi-squared test, which was resolved by using a negative binomial distribution.

## 3. Results

*Neoleucopis kartliana* was present at all sites (Figure 1) in this study; however, its abundances varied widely between sites and over the season.

The fly was observed in every subadult developmental stage on the branches (eggs, larvae, puparia). Eggs were usually located inside or close to the cotton-like wax produced by GPS. Larvae were spotted either inside the ovisacs of GPS or close to other developmental stages (first, second, and third instar nymphs and adults). Puparia were found either inside the wax of GPS or in bark crevices, without the presence of GPS being necessary.

The data from Kedrinos Lofos (Thessaloniki) (*n* = 1124 individuals) suggested that, unlike its univoltine prey, the silver fly has three generations per year in northern Greece (Figure 2). The fly’s eggs were found during all developmental stages of GPS. However, the graphical analysis of *N. kartliana*’s generations was based on the relative abundance of larvae and puparia only, as they are greater in size and could therefore be more easily detected compared to the eggs. *N**eoleucopis kartliana* larvae were observed feeding on all developmental stages of GPS. In the first *N. kartliana* generation, larvae (young and mature) preyed mostly on GPS eggs and adults; in the second fly generation, larvae preyed on the first-instar nymphs of GPS; while in the third fly generation, larvae preyed on the second- and third-instar nymphs of GPS (Figure 2). Although *N. kartliana* larvae did not extensively prey on the third instar nymphs of GPS due to overwintering as puparia, early emerging larvae of the subsequent fly generation (first) were found preying on third instar nymphs of GPS and remained attached to their prey during the scale’s ecdysis.

A total of 6031 *N. kartliana* adults were sexed to estimate the overall sex ratio. With 50.8%:49.2% males:females, the sex ratio was almost 1:1, and no apparent difference in the sequence of emergence was observed between the sexes.

Data on the artificial food preference of adults could not be retrieved from this test, because the flies tended to frequently roam inside the cage. However, whenever flies were observed feeding, they were found on only two of the media provided. Adults introduced into the cage containing the different artificial food sources were observed to mainly feed on dry yeast diluted with sugar and, to a lesser extent, on the mixture of honey and yeast. When introduced into the cage, the flies roamed, inspecting the various artificial food sources. However, most flies soon gathered, attached their mouthparts, and fed only on the food sources mentioned above. Adults survived approximately two weeks in captivity with a sole providence of artificial food sources; however, this should not be considered as the fly’s lifespan, as it was not estimated individually, but rather in groups of 15 ± 5 adults.

There was a significant effect of female age on the number of mature eggs found in the ovaries (χ^2^ = 112.77; df = 1; *p* < 0.0001). Dissections showed that within 24 h of emergence, females had either zero (*n* = 21) or one to two (*n* = 4) eggs in their ovaries. However, until the eighth day after emergence, eggs matured in an exponential manner (Figure 3), and a mean of 25.7 eggs was found, with a maximum of 79 mature eggs found in one female.

## 4. Discussion

Results from this study support the proposed use of the predatory fly *N. kartliana* as a classical biological control agent to minimize the impact and spread of *M. hellenica* in Australia [17]. The silver fly has a high intrinsic growth rate, allowing it to undergo three generations per year in northern Greece, while its host is univoltine. It seems to prey indiscriminately on every developmental stage of the scale and was found to be present in all sites studied in northern Greece. Furthermore, potential fecundity of the fly was found to increase exponentially in the first eight days after emergence with an average of 25.7 eggs and females holding up to 79 eggs in their ovaries.

*N. kartliana* appears to have at least three generations per year, confirming the observations of Gaimari et al. [32], who suggested that *N. kartliana* has two to three generations annually. Additionally, other species of the genus *Neoleucopis* have been described to be at least bivoltine, e.g., *N. pinicola* [37] and *N. atratula* [38,39]. Multivoltinism is an attribute that may considerably increase the chances of adaptation to novel environments because it imparts the capability of surviving and reproducing under various environmental conditions [40]. In a study on introduced biological control agents, Crawley et al. [41] found that insects with the highest intrinsic growth rates that typically also had other characteristics of r-selected species (smaller body size and faster maturity resulting in several generations per year) were more likely to establish successfully than those with a slower growth rate. Accordingly, Hokkanen and Sailer [42] suggested that there is a positive correlation between success in biological control and the agent’s power of increase over that of the prey, supporting that, in general, successful natural enemies have two generations for every host generation. An example supporting this theory is the parasitoid *Aphytis melinus* (DeBach) (Hymenoptera, Aphelinidae), which was successfully used as a biological control agent against the California red scale, *Aonidiella aurantii* (Maskell) (Hemiptera, Diaspididae), a worldwide pest of citrus [43]. According to Murdoch et al. [43], apart from prey specificity, another key feature leading to the success with this species appears to be the rapid development of *A. melinus* compared to the development of the pest, since the parasitoid has three generations for each scale generation. However, recent research suggests that, when considering the multivoltinism of a biological control agent in a more holistic context of biological control programs, agent-related life-history traits play a rather minor role, compared to those related to how and when agents are released [44]. Nevertheless, we show here that the three generations of *N. kartliana* allow this predator to feed on all life stages of the scale, which would maximize its impact on *M. hellenica* populations. This finding confirms those of previous studies on the feeding habits of Chamaemyiidae. For example, Satar et al. listed six *Leucopis* species that were observed preying on several developmental stages of aphids in Turkey [45].

We found that adult *N. kartliana* males and females emerged simultaneously, similar to *Leucopis argenticollis* and *L. piniperda* [46] (later both assigned to a new genus, *Leucotaraxis* [47]). Additionally, our data suggest that *N. kartliana* adults follow the Fisherian sex ratio (1:1) [48], which was also found for *N. pinicola* following laboratory rearing of field-collected puparia [37].

*Neoleucopis kartliana* adults survived for two weeks in captivity, successfully feeding solely on artificial food sources that consisted of water and a mixture dry yeast and sugar, as was also offered successfully to other silver flies [45]. The adults rejected or showed little interest in the alternative artificial food sources (pine honey, pine honey with dry yeast, milk, and a mixture of raw yeast and sugar). Flies generally require both sugar and protein to fully develop their reproductive systems and produce eggs, and different sources of protein can have various effects on longevity and fertility [12,17]. Although the traditional protein source used for dipteran species is milk powder, yeast could replace milk powder without a considerable loss of viability or egg production [12]. Chamaemyiidae flies are known to feed on honeydew produced by their host [49], which is a sugar-rich secretion of aphids and scale insects [50]. Because it is difficult and laborious to keep honeydew-producing scale insects alive on their host plants (especially when they are trees), replacing the natural food source with artificial ones can increase the efficiency of adult rearing. Gaimari and Turner [1] suggested the use of a mixture of honey and yeast as an artificial diet for adult *Leucopis* spp. While a mixture of sugar and yeast was used as an artificial diet for the *N. kartliana* adults in this study, our results do not contradict those of Gaimari and Turner, since *N. kartliana* adults did feed on the diet suggested by the authors but ultimately preferred the mixture of yeast and sugar. The comparative impact of the different food sources on the longevity and fecundity of *N. kartliana* remains to be studied.

*Neoleucopis kartliana* is oviparous, corresponding with most dipteran species [51]. Our results showed that females emerge with no or very few (one to two) mature eggs in their ovaries. The few eggs that were found in some freshly emerged females may have been developed in the few hours between emergence and dissection. Clearly, most of *N. kartliana*’s eggs mature after its emergence, making it a strongly synovigenic species (i.e., producing eggs throughout its adult life), which are typically relatively long-lived and dependent on external food sources, as shown for parasitoids [52]. The presence of mature eggs in an ovary is considered as the definitive characteristic for female sexual maturity [13], suggesting that there should only be a very short or no pre-mating period after emergence of females. This study showed that, over the span of 8 days, egg load increased exponentially to a mean of 25.7 eggs and a maximum of 79 mature eggs per female. Possibly, the eighth day after the emergence of *N. kartliana* is the transition point between sexually immature and mature female flies, similar to another Chamaemyiid species, *Leucopis palumbii* Rodani, that reaches sexual maturity at 8–10 days after emergence [53]. While this illustrates a promising intergenerational population growth rate, the silver fly’s fecundity in nature has not yet been estimated, and it remains unclear if more eggs can be produced over a female’s lifetime and if factors such as mating, food source, or environmental temperature affect egg production and, ultimately, realize the fertility of *N. kartliana*. Nevertheless, the maximum of 79 eggs produced by female flies in captivity signifies prospects for an increase in egg production above the current mean of 25.7 eggs if adjusted methods for rearing are practiced.

## 5. Conclusions

Phenological observations revealed that *N. kartliana* has at least three generations per year in northern Greece and is preying indiscriminately on every developmental stage of the univoltine GPS. According to laboratory observations, the silver fly is oviparous and produces eggs without mating occurrence, and can survive for about two weeks in captivity when provided with artificial food sources consisting of water, sugar, and dry yeast. Adult females emerge with no or very few mature eggs in their ovaries, after which egg production increases exponentially until at least the eighth day after emergence. This investigation of the silver fly’s life-history traits helps to better understand its biology and contribute to its evaluation as a classical biological control agent of the invasive GPS in Australia. However, research on the fly’s prey specificity, mating behavior, rearing, as well as definite lifespan and egg load throughout its lifespan remains to be conducted in order to further understand its behavioral ecology and safe use as a biological control agent and to optimize its chances of establishment, as other Chamaemyiid biological control agents have failed to establish in Australia [54].

## Figures and Tables

**Figure 1 insects-13-00143-f001:**
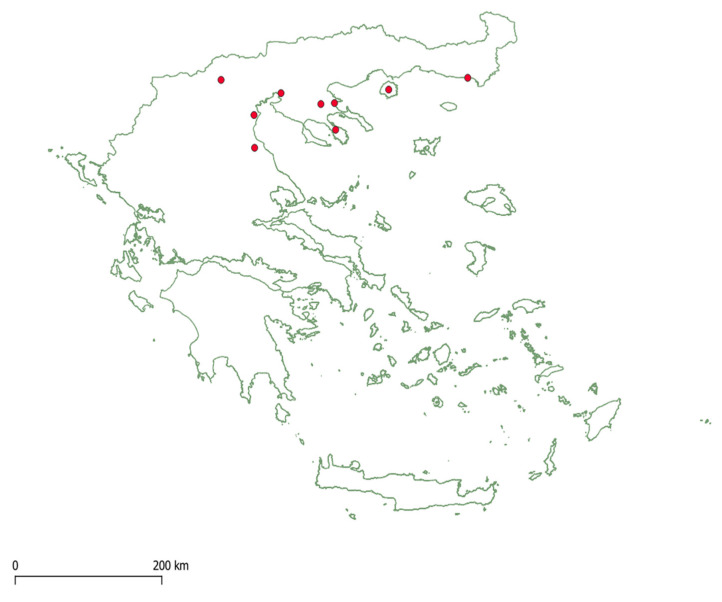
Sampling locations for *Neoleucopis kartliana* in northern Greece.

**Figure 2 insects-13-00143-f002:**
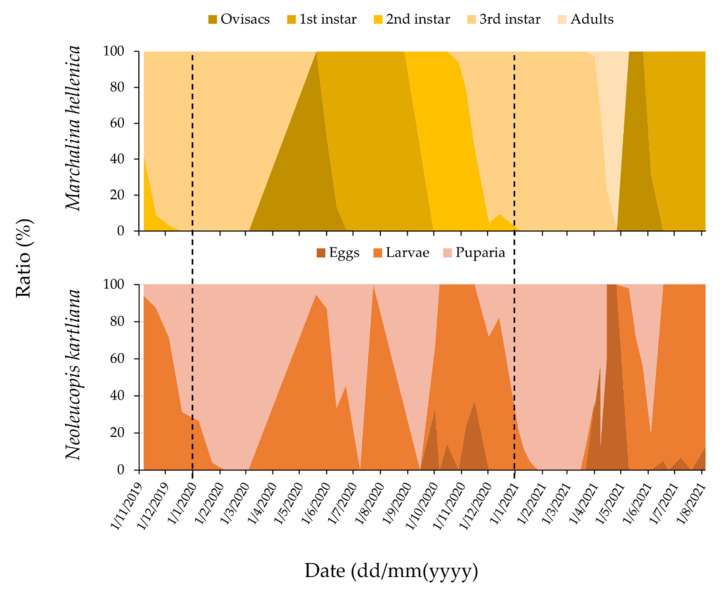
Ratio (%) of different developmental life stages of (upper panel) *Marchalina hellenica* and (lower panel) *Neoleucopis kartliana* in Kedrinos Lofos (Thessaloniki) between November 2019 and August 2021. The area between the two dashed lines is one full year (2020), in which *M. hellenica* underwent one full generation and *N. kartliana* underwent three.

**Figure 3 insects-13-00143-f003:**
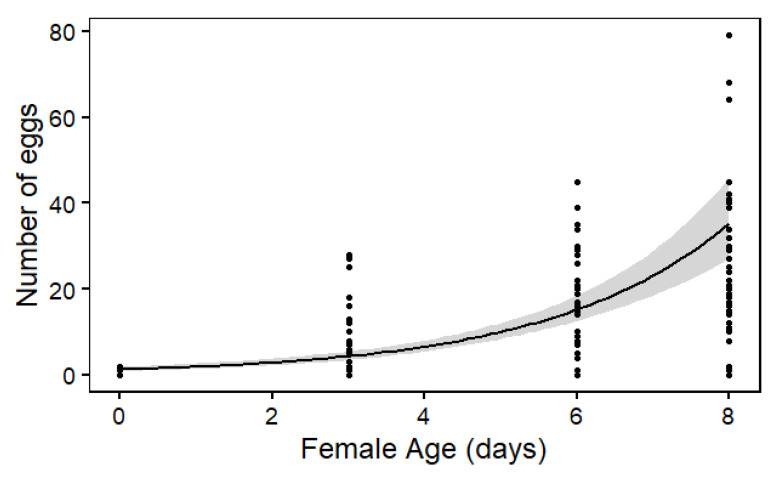
Number of mature eggs in ovaries of *Neoleucopis kartliana* at different ages. The regression line indicates the predictions of the negative binomial generalized linear model that are back-transformed from the log scale. The gray area around the line shows 95% confidence intervals.

## Data Availability

Data are contained within the article.

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
