# Peer review of "Phenology and Potential Fecundity of Neoleucopis kartliana in Greece"

_insects, 2022, doi:10.3390/insects13020143_

Round 1

Reviewer 1 Report

This paper presents data on a potential biocontrol predator of giant pine scale, a new invasive species in Australia. The data is from Greece, within the native range of the silver fly predator (Neoleucopis kartliana) and its prey (Marchalina hellenica). The data presented is relevant to evaluation of a potential biocontrol and will be of great interest for Australians in the field. There are impressive sample sizes for sex ratios and female egg load. The statistical analysis of eggs is well done. Some more information should be provided to improve the phenology portion.

The citation of literature in the Introduction and Discussion is poorly done and full of errors. In particular, the authors fail to cite primary sources or they cite misleading or erroneous information. The reference list formatting is rough and unfinished.

Line comments

49-50:  citation of papers 4-6 for the statement “majority of chamaemyiids have 1-3 generations” is not appropriate. These 3 papers deal with very few of the world’s chamaemyiids. Look for a better primary source such as a taxonomic review.

53: citation of paper 7 for “descend from the tree and pupate in the soil” is false. In that paper, the insect which pupates in soil is a beetle, not a chamaemyiid.

55-56:  what is the genetic factor in paper 8? Only one paper to support this long list of factors?  Seems more likely that in paper 8, they cited other papers for some of those factors.

57-58: citation of 10-11 for “throughout the world” are not primary sources. In those papers you will find lists of several sources that support the statement.

86-91: in this section the authors speak disrespectfully about the paper that laid the groundwork for their own investigation. Paper 22 being titled, “Notes on..” makes me think it was not intended as an exhaustive investigation of this species but rather as a starting point. Indeed, the authors cite it again on lines 112 and 128 to support their methodology. Instead of casting negativity on papers others have written, simply say what they did, then say how your paper fills a need.

103-105: are these sites very similar in their characteristics? Are they forest, rural, urban, and what about elevation? Was there a certain rationale for choosing sites? It looks like the data for phenology was pooled across all the sites. So, you need to convince the reader that these sites were on the same phenological “baseline” and that the data aren’t skewed by one site that is ahead or behind the others in phenology. I would like to see a table of how many flies were collected broken down by date and site, and also how many adult females were used from each site for the fecundity study.

109: typo “of with”

114: are your vouchers going to be accessioned in a museum or insect collection? If so, give location.

118: remove “larvae,” this is a typo.

120: is this a standard brand-name climate chamber, or something custom-made? I haven’t heard of one with dimmable lights before.

129: please add information about how many flies per cage and how many food treatments per cage. Also, is there a citation for these flies feeding on GPS honeydew, or is that something you observed directly? Please clarify if this is known or just an assumption.

135: it is more clear to say “dry yeast diluted with sugar” rather than “diluted dry yeast with sugar.”

135: treatment 6 sounds like it was really a variety of treatments, how were these split among the cages?

139: how was food acceptance evaluated? How many times a day were flies observed, and at what time of day were they observed?

141: you can’t be sure that the females are virgin, since you sometimes didn’t check the cage every day and they could have mated in the cage. Was there a procedure during collection to attract the flies up out of the foliage – such as bright lights? Flies might tend to stay on the foliage and not be easily collected for a while. To assure virginity, you would need to isolate puparia and rear them individually.

But maybe it’s not very important that they be virgin for your assessment of eggs. You could just say females.

150: I believe the word “stripe” is meant instead of “strip.”

161: as stated earlier a table with sites and dates would be helpful.

164: misspelled Neoleucopis in the figure caption

168, 173, &etc.: The correct term for the group Cyclorrhapha is puparium, not pupa. The pupa forms within the puparium and is not visible. See Fraenkel, G. and G. Bashkaran (1973) Pupariation and pupation in Cyclorrhaphous flies (Diptera): terminology and interpretation.  Annals of the Entomological Society of America 66(2):418-422.

177: Figure 2. Again, is this just one site (Kedrinos Lofos), or pooled across sites? This is important to know. Provide sample sizes for the flies. I understand that you looked at a minimum of 100 GPS per sampling but was that at one, or all sites?

189: can you say any more about lifespan? An average or a range? “Approximately” is not very descriptive, it might represent a maximum if you didn’t count dead flies along the way.

197: for clarity/emphasis you might want to say “number of mature eggs” in the figure caption.

202: what other steps need to be taken to evaluate a predator for Australia? Specificity testing? Anything else? See also lines 273-275.

208-209: You may want to compare to: Benelli G. et al., First quantification of courtship behavior in a silver fly, Leucopis palumbii (Diptera: Chamaemyiidae): Role of visual, olfactory and tactile cues. J. Insect Beha. (2014) 27:462-477. In this paper, they state that the species reaches maturity after 8-10 days, although I cannot locate any primary data for that.

212-213: Why not cite paper 35 here?

235-237: major error in citing paper 33. The flies in that paper feed on Adelges tsugae in the northwest of North America. This paper also talks about a beetle. The flies feed mainly on adelgid eggs in both generations of the adelgid, while the beetle feeds on all life stages but only one generation. Together, the 3-predator complex feeds on the entire life cycle of Adelges tsugae. I don’t even think the paper is relevant to the statement that you wish to support, but I believe you could find some.

If you are going to cite papers published about Leucopis argenticollis and Leucopis piniperda (such as papers 4, 5, 7, 10, 11), you should also cite this paper in which the genus name was changed to Leucotaraxis: Gaimari, S. D. and N. P. Havill (2021) A new genus of Chamaemyiidae (Diptera: Lauxanioidea) predaceous on Adelgidae (Hemiptera), with a key to chamaemyiid species associated with Pinaceae-feeding Sternorrhyncha. Zootaxa 5067(1):001-039.

251: it is my understanding that honeydew is generally NOT protein-rich, in fact is extremely lacking in protein. It may contain certain amino acids but certainly not a lot of protein. It is the waste left over after the insect has extracted the useful protein. A quick search of paper 38 (cited) reveals no mention of protein or amino acids in the entire paper.

Much more difficult than finding a food source for the adult is creating an artificial diet for the larvae. The need to rear Chamaemyiids with their prey’s host plant makes them difficult to rear in mass.

273-275: another follow-up study could investigate the number of GPS that each larva can eat over the life stage. And specificity testing in choice and no-choice feeding trials. If these have been done, you should cite them, if you are trying to build a case for these insects as biocontrol agents.

Another question is what is the history of introducing Chamaemyiids in Australia for other pests? Have there been any established, and if so, could they potentially feed on GPS? There is a long history of attempts to use chamaemyiids around the world.

Reviewer 2 Report

The study shows the phenology of an invasive pest, Marchalina hellenica (Hemiptera, Margarodidae) and the natural enemy, Neoleucopis kartliana (Diptera, Chamaemyiidae), in their native area Greece. No information to compare the phenology of the pest, GPS, in the invasive area (Australia) is mentioned either in the introduction or the discussion sections.

Figure 2. I don’t think that the relative abundance is enough information to show in a manuscript for this journal. Number of different stages of the pest and the natural enemy per twigs and/or branches would be more valuable information.

To  study the acceptance of artificial food sources as substitute of GPS honeydew this manuscript states results about six different artificial diets; 1) water provided through a constantly soaked cloth strip laid loosely on a vial, 2) pine honey, 3) pine honey mixed with dry yeast diluted in water (2ml:1gr:100ml), and 4) water-diluted pasteurized milk (50:50) provided through soaked cotton laying on a petri dish (8.5 cm diam-134 eter), 5) diluted dry yeast with sugar, 6) raw moist yeast mixed with sugar were both provided in different rates (5%-50%). No results are shown in result section about the diets. The authors discuss about this results in the discussion section without showing any result. I would like to see some numbers about preference, mortality, egg load or feeding behavior to, somehow, show that the results are countable.

To calculate the egg load of the natural enemy at different times, the authors use the data from two different diets. Why do they not compare this data between diets?

Line 109: Please provide the total number of samples that were examined.

Reviewer 3 Report

I think this is an excellent step towards using this species as a biocontrol agent where the scale species is a pest. Very interesting new information. I have only a few comments on the attached marked-up manuscript.

Round 2

Reviewer 1 Report

Overall this revision is much improved and my comments were addressed. The methods are clarified and the citations are better.

However, the revision of the paragraph in lines 86-89 is still not appropriate.

In my original review I suggested that the authors should, “Instead of casting negativity on papers others have written, simply say what they did, then say how your paper fills a need.” This revision removed any useful information about Gaimari et al. 2007 and replaced it with unnecessary praise words like detailed, unique, novel. PLEASE simply give a summary of the findings of the 2007 paper. Tell us more information about this species! The purpose of the introduction is to provide the background information.
